# REINFORCEGEN: HYBRID SKILL POLICIES WITH AUTOMATED DATA GENERATION AND REINFORCEMENT LEARNING

## ABSTRACT

Long-horizon manipulation has been a long-standing challenge in the robotics community. We propose ReinforceGen, a system that combines task decomposition, data generation, imitation learning, and motion planning to form an initial solution, and improves each component through reinforcement-learning-based fine-tuning. ReinforceGen first segments the task into multiple localized skills, which are connected through motion planning. The skills and motion planning targets are trained with imitation learning on a dataset generated from 10 human demonstrations, and then fine-tuned through online adaptation and reinforcement learning. When benchmarked on the Robosuite dataset, ReinforceGen reaches 80% success rate on all tasks with visuomotor controls in the highest reset range setting. Additional ablation studies show that our fine-tuning approaches contributes to an 89% average performance increase. More results and videos available in https://sites.google.com/view/reinforcegen-iclr26.

## 1 INTRODUCTION

Imitation Learning (IL) from demonstrations is an effective approach for agents to complete tasks without environmental guidance. In long-horizon tasks, collecting demonstrations can be expensive, and the trained agent is more likely to deviate from the demonstrations to out-of-distribution states. Reinforcement Learning (RL) leverages random exploration, incorporating environmental feedback through rewards. However, long horizons exacerbate the exploration challenge, especially when the reward signals are also sparse. In the context of robot learning, collecting demonstrations for IL is often time-consuming and expensive, as it typically requires a teleoperation platform and coordinating with human operators. Furthermore, the solution quality and data coverage of the demonstrations are critical, as they directly impact the agent's performance when used in methods such as Behavior Cloning (BC) (Mandlekar et al., 2021).

One promising solution to combat demonstration insufficiency is to augment the dataset through synthetic data generation. In robotic manipulation tasks, a thread of work (Mandlekar et al., 2023b; Garrett et al., 2024; Jiang et al., 2025) focuses on object-centric data generation through demonstration adaptation. Other approaches (McDonald & Hadfield-Menell, 2022; Dalal et al., 2023) use use Task and Motion Planning (TAMP) (Garrett et al., 2021) to generate demonstrations. An alternative strategy is to hierarchically divide the task into consecutive stages with easier-to-solve subgoals (Mandlekar et al., 2023a; Garrett et al., 2024; Zhou et al., 2024). In most manipulation tasks, only a small fraction of robot execution requires high-precision movements, for example, only the contact-rich segments. Thus, these approaches concentrate the demo collection effort at the precision-intensive skill segments and connect segments using planning, ultimately improving demo sample efficiency.

Still, these demonstration generation methods are open-loop and rely solely on offline data. As a result, IL agents trained with the generated data are still bottlenecked by the quality of the source demonstrations. To combat this, we propose ReinforceGen, a framework that improves hierarchical data generation by incorporating online exploration and environmental feedback using RL. ReinforceGen trains a hybrid BC agent with object-centric data generation as its base policy. It then combines distillation, causal inference, and RL to improve the base agent with online data, as well as real-time adaptation from environment feedback during deployment. We demonstrate that ReinforceGen

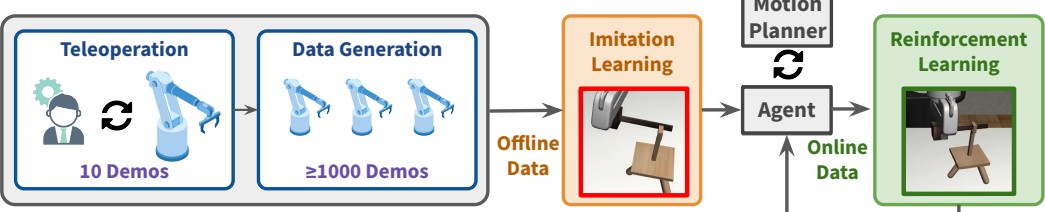

**Figure 1:** ReinforceGen first creates an offline dataset by synthetic data generation from a small set of source human demonstrations. The dataset is then used to train a hybrid imitation learning agent that alternates between moving to a predicted waypoint using a motion planner and directly controlling the robot using a learned policy. Finally, ReinforceGen uses reinforcement learning to fine-tune the agent with online environment interactions.

produces high-performance hybrid data generators and show that ReinforceGen-generated data can also be used to train end-to-end visuomotor imitation policies.

**The contributions of this paper are the following.**

We propose ReinforceGen, an automated demonstration generation system that integrates planning, behavior cloning, and reinforcement learning to train policies that robustly solve long-horizon and contact-rich tasks using only a handful of human demonstrations.

Through using localized reinforcement learning, ReinforceGen is able to explore and thus go beyond existing automated demonstration systems, which are fundamentally bounded by the performance of the demonstrator, and learn more successful and efficient behaviors.

At deployment time, ReinforceGen executes a hybrid control strategy that alternates between motion planning and fine-tuned skill policy segments, where the use of planning also at deployment reduces the generalization burden of learning, resulting in higher success rates.

We evaluate ReinforceGen on multi-stage contact-rich manipulation tasks. ReinforceGen reaches an **80%** success rate, almost doubling the success rate of the prior state-of-the-art (Garrett et al., 2024).

Finally, we train proficient end-to-end imitation agents with ReinforceGen.

## 2   RELATED WORK

**Imitation and reinforcement learning for robotics.** Imitation Learning (IL) from human demonstrations has been a pillar in robot learning, benchmarking significant results in both simulations and real-world applications (Florence et al., 2022; Chi et al., 2023; Shafiullah et al., 2022; Zhao et al., 2023; Kim et al., 2024). Reinforcement Learning (RL) is another promising approach to solve robotic problems with even real-world successes (Tang et al., 2025; Wu et al., 2022; Yu et al., 2023). However, without engineered rewards, RL often struggles with exploration and can even outright fail (Zhou et al., 2024). Similar to our approach, Nair et al. (2017); Johannink et al. (2018); Zhou et al. (2024) investigate combining the strength of IL and RL, using the IL policy as a warm start for RL for exploration and using RL to fine-tune the IL agent in out-of-distribution states. However, ReinforceGen also leverages motion planning to decompose control into smaller skill-learning subproblems, dramatically boosting policy success rates.

**Automated data generation.** McDonald & Hadfield-Menell (2022); Dalal et al. (2023) use Task and Motion Planning (TAMP) (Garrett et al., 2021) to generate demonstrations for imitation learning; however, these systems focus on prehensile manipulation due to the difficulty of modeling contact-rich skills. Dalal et al. (2024) leverage Large Language Models (LLMs) for task planning in place of full planning model. MimicGen (Mandlekar et al., 2023b), SkillMimicGen (Garrett et al., 2024), DexMimicGen (Jiang et al., 2025), DemoGen(Xue et al., 2025), CP-Gen (Lin et al., 2025), and MoMaGen (Li et al.). automatically bootstrap a dataset of demonstrations from a few annotated human source demonstrations through pose adaptation and trial-and-error execution. A key drawback is the the dataset has limited diversity because to each demonstration is derived from the same small set. In ReinforceGen, we directly address this by using reinforcement learning to explore beyond these demonstrations in search of more successful and less costly behaviors.

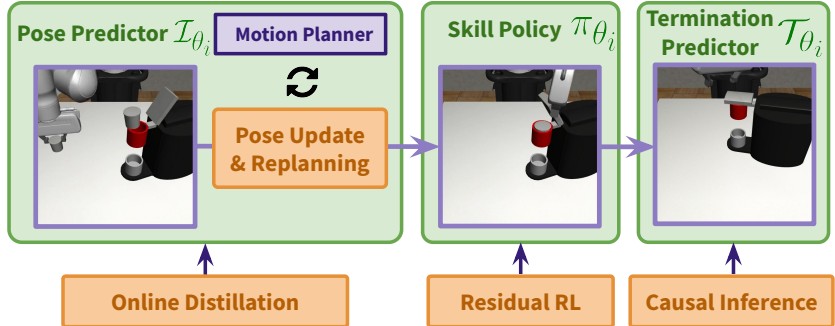

**Figure 2:** The three main components of a ReinforceGen stage. The pose predictor $\mathcal{I}_{\theta_i}$ predicts the target end-effector pose and updates the motion planner in real-time. After reaching the destination, the skill policy $\pi_{\theta_i}$ takes control to complete the stage goal, which is determined by the termination predictor $\mathcal{T}_{\theta_i}$. All three components are first imitated from a generated dataset, then fine-tuned with online data.

**Hybrid planning and learning systems.** Several approaches integrate learned policies into TAMP systems. Wang et al. (2021) engineered open-loop parameterized skill policies and learned successful skill parameters using Gaussian Process regression. NOD-TAMP (Cheng et al., 2024) used Neural Object Descriptors (NODs) to warp human skill demonstrations to new object geometries and deploy them open-loop within TAMP. HITL-TAMP (Mandlekar et al., 2023a) learned closed-loop visuomotor policies in place of open-loop skills using Behavior Cloning (BC). The closest to our approach is SPIRE (Zhou et al., 2024), which goes beyond HITL-TAMP by using Reinforcement Learning (RL) to fine-tune initiation-learned skills, improving their success rates and decreasing execution costs. In order to leverage TAMP, all these approaches assume access to an engineering planning model that specifies the preconditions and effects of each skill. In contrast, ReinforceGen learns and fine-tunes initiation and termination models that implicitly encode these dynamics without prior information.

## 3 PRELIMINARIES

We first describe our problem class (Section 3.1) and overview prior work that we build on regarding demonstration adaptation (Section 3.2) and interleaved imitation learning and planning (Section 3.3).

### 3.1 PROBLEM FORMULATION

We model manipulation tasks as Partially Observable Markov Decision Processes (POMDPs) where $\mathcal{S}$ is the state space, $\mathcal{O}$ is the observation space, $\mathcal{A}$ is the action space, and $\mathcal{T} \subseteq \mathcal{S}$ is a set of terminal states. The reward function is deducted from reaching a terminal state $r(s) := [s \in \mathcal{T}]$. We are interested in producing a policy $\pi : \mathcal{O} \to \mathcal{A}$ that controls the system from initial state $s_0 \in \mathcal{S}$ to a terminal state $s_T \in \mathcal{T}$ while minimizing the number steps taken.

### 3.2 OBJECT-CENTRIC DATA GENERATION

ReinforceGen builds on the data collection methodology introduced in MimicGen (Mandlekar et al., 2023b). MimicGen automatically generates data by transforming and replaying human demonstrations for manipulation tasks. It assumes the environment contains a set of manipulable objects $M = \{O_1, \dots, O_m\}$, and their poses comprise a component of states $s \in \mathcal{S}$. MimicGen divides the task into multiple contiguous object-centric subtask segments, where each segment $i$ is associated with a fixed reference object $R_i \in M$. Let $T_B^A \in \mathrm{SE}(3)$ be the homogeneous transformation matrix representing object $A$'s pose in the frame of object $B$, where $A, B \in M$, and let $W$ be the world frame. For a skill segment involving object $A$, the source trajectories of end-effector poses $\{T_W^{A_t}\}$ are parsed into the frame of the corresponding reference objects $T_{R_i}^{A_t} \leftarrow (T_W^{R_i})^{-1} T_W^{A_t}$. To generate new data in a new environment instantiation, segments of a sampled source trajectory are adapted according to the current reference object $R_i'$ pose: $T_W^{A_t'} \leftarrow T_W^{R_i'} T_{R_i}^{A_t}$. We then extract the delta pose actions from the transformed set of poses and execute the action sequence with the original gripper actions. After execution, successful trajectories are be retained and added to the dataset.

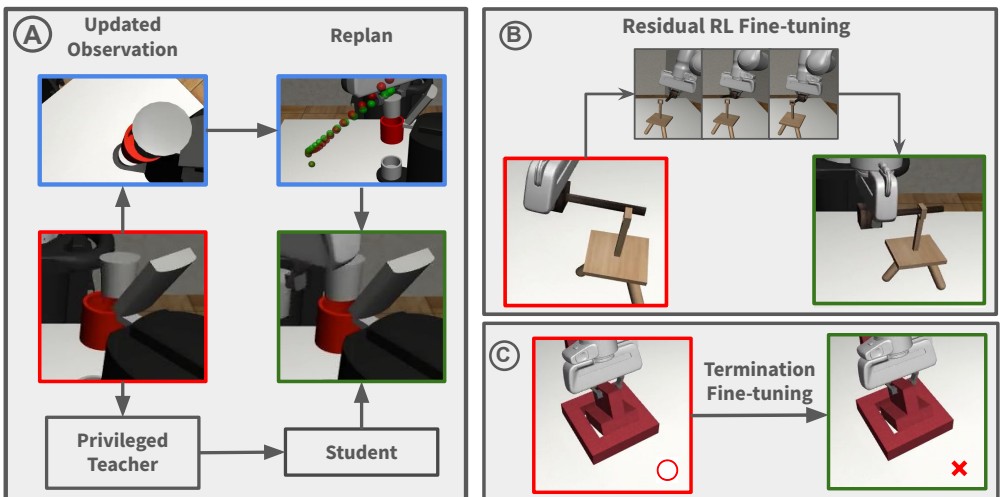

**Figure 3:** Depiction of how we fine-tune the three components. (A) The pose predictor is fine-tuned towards a privileged teacher; During execution, it constantly updates its prediction based on new observations and reroutes when the deviation is too large. (B) The skill policy is fine-tuned through residual reinforcement learning. (C) We purge the false-positive predictions from the termination predictor.

## 3.3 HYBRID SKILL POLICY

ReinforceGen adopts the bilevel decomposition from the Hybrid Skill Policy (HSP) framework (Garrett et al., 2024) to reduce the complexity of long-horizon tasks, namely the stage decomposition and the skill decomposition. The stage decomposition separates the task into sequential stages, each with a different sub-goal that progresses towards the task goal. The skill decomposition splits the task into precision-intensive skill segments that are directly responsible for task goal or sub-goal completion. This decomposition allows us to concentrate learning effort on the skill segments.

An HSP decomposes the task into $n$ stages, where each stage is comprised of a connect segment followed by a skill segment. In the connect segment, the robot moves from its current position to the initiation position of the skill segment. Following the *options* framework (Stolle & Precup, 2002), the $i$-th skill in the skill sequence $\psi_i := \langle \mathcal{I}_i, \pi_i, \mathcal{T}_i \rangle$ consists of an initiation end-effector pose condition $\mathcal{I}_i \subseteq \mathrm{SE}(3)$, a skill policy $\pi_i : \mathcal{O} \to \mathcal{A}$, and a termination condition $\mathcal{T}_i \subseteq \mathcal{S}$. When applying this formation to object-centric data generation (Sec. 3.2), we assume that the skill segment in each stage uses a fixed reference object $R_i \in M$. Similarly, the initiation end-effector pose can also be generated in an object-centric fashion by transforming the target pose $E_0$ in the source trajectory in the frame of the original reference object to the current: $T_W^{E_0'} \leftarrow T_W^{R_i'} T_{R_i}^{E_0}$, as in Garrett et al. (2024). We describe how to learn a parameterized HSP agent in the next paragraph.

Formally, a *Hybrid Skill Policy* (HSP) (Garrett et al., 2024) agent is defined by a sequence of parameterized skills $[\psi_{\theta_1}, ..., \psi_{\theta_n}]$. Each skill $\psi_{\theta_i} := \langle \mathcal{I}_{\theta_i}, \pi_{\theta_i}, \mathcal{T}_{\theta_i} \rangle$ is comprised of an *initiation predictor* $\mathcal{I}_{\theta_i} : \mathcal{O} \to \mathrm{SE}(3)$ that predicts a skill start pose in $\mathcal{I}_i$ based on the current observation, a parameterized *skill policy* $\pi_{\theta_i} : \mathcal{O} \to \mathcal{A}$, and a *termination classifier* $\mathcal{T}_{\theta_i} : \mathcal{O} \to \{0, 1\}$ that determines whether the current state is a terminal state in $\mathcal{T}_i$. The parameters $\{\theta_i\}$ are learned through behavior cloning, maximizing the likelihood of producing the generated data: $P(E_0' = \mathcal{I}_{\theta_i}(o_0))$, $P(A_t' = \pi_{\theta_i}(o_t))$, and $P([s_t \in \mathcal{T}_i] = \mathcal{T}_{\theta_i}(o_t))$. At each stage, an HSP predicts a starting pose with $\mathcal{I}_{\theta_i}(o_0)$, moves to the pose with motion planning, and executes the skill policy $\pi_{\theta_i}$ until $\mathcal{T}_{\theta_i}$ predicts termination. As in prior work (Mandlekar et al., 2023b; Garrett et al., 2024), we assume the stage sequence along with each reference object $R_i$ are annotated by a human per task.

## 4 REINFORCEGEN

Existing object-centric data generation methods (Mandlekar et al., 2023b; Garrett et al., 2024; Mandlekar et al., 2023a) have been shown to be capable of generating high-quality trajectory datasets that derive performant policies in long-horizon manipulation tasks from a small set of human demonstrations. Despite their impressive performances, these methods are still bounded by the quality of the demonstrations since they primarily replay transformed source trajectories

or directly run Imitation Learning (IL) on the collected demonstrations, without exploring new behaviors. ReinforceGen presents a solution to improve data generation quality substantially by integrating a trained HSP imitation learning policy (Sec. 3.3) and online environment feedback into the data generation workflow, allowing the policy, and ultimately the generated data, to improve over time via exploration and exploitation. Specifically, we propose fine-tuning pipelines for individual parameterized components of an HSP with reinforcement learning, namely initiation pose predictor $\mathcal{I}_{\theta_i}$ (Sec. 4.1), skill policy $\pi_{\theta_i}$ (Sec. 4.4), and termination predictor $\mathcal{T}_{\theta_i}$ (Sec 4.3). With the fine-tuned components, we execute these skill policies in a hybrid policy that alternates between motion planning and the skills (Sec. 4.4). Finally, we demonstrate the ability to distill the hybrid policy to a fully end-to-end policy that does not rely on a motion planner (Sec. 4.5).

## 4.1 INITIATION POSE PREDICTION

The initiation pose predictor $\mathcal{I}_{\theta_i}$ proposes a target pose for the motion planner to reach before handing the control off to the skill policy. It plays a critical role in the HSP framework since it directly dictates the distribution of the skill policy. Even small errors can lead to out-of-distribution states and cause severe performance degradation. To illustrate this, we artificially add noise to the initiation pose with increasing scales and plot the success rate of the following skill policy against it. As shown in Fig. 4, the success rate drops sharply as the prediction error increases. In practice, such errors can often occur, especially when the agent has imperfect perceptions. To mitigate this, we propose two methods that utilize online interactions during deployment and in training, respectively.

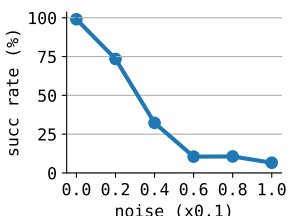

**Figure 4:** Success rate drops sharply as the pose target noise level increases in the second stage of *Nut Assembly*. See App. E.1 for more details.

**Real-time replanning.** As the robot arm approaches the target, more information can be gathered to predict a better initiation pose. A typical scenario is that a robot equipped with a wrist camera can only capture the target location when the end-effector is close to it. Therefore, instead of running the pose predictor once, we run it at every timestep of the connect phase, while executing a motion plan. When the difference between the newly predicted pose and the previously planned target exceeds a threshold, we then restart motion planning with the updated pose.

$$
T_W^{E_0'} \leftarrow
\begin{cases}
T_W^{E_0'} & \text{if } \text{dist}\!\left(T_W^{E_0'}, \mathcal{T}_{\theta_i}(o_t)\right) \leq \epsilon_{\text{pose}}, \\
\mathcal{I}_{\theta_i}(o_t) & \text{otherwise.}
\end{cases}
\tag{1}
$$

We use the maximum between Euclidean distance and rotational difference as our pose difference measure.

**Student predictor from a privileged teacher.** The replanning approach stresses the importance of continuously making predictions during motion planning. However, when using a suboptimal predictor, the motion planning trajectory will inevitably encounter out-of-distribution states, thereby hindering prediction accuracy. Fortunately, during training, we have access to privileged object state information that allows us to use an accurate teacher predictor. Specifically, the privileged predictor selects one of the source demos and transforms the target pose in the demo according to the current object state, similar to the data generation procedure described in Sec. 3.2. We denote the teacher predictor $\mathcal{I}^{\text{Priv}}$, and the HSP variant that uses this predictor as *HSP-Priv* (equivalent to HSP-Class in Garrett et al. (2024)). Given the teacher predictor, we can distill our predictor with:

$$
\mathcal{L}_{\text{pose}}(\theta_i) = \mathbb{E}_{\tau_{\text{connect}} \sim \overline{\mathcal{I}}}\left[\frac{1}{2}\left\|\mathcal{I}_{\theta_i}(o_t) - \mathcal{I}^{\text{Priv}}(s_t)\right\|_2^2\right],
\tag{2}
$$

where $\overline{\mathcal{I}}$ is the predictor used to generate the trajectories. We start with $\overline{\mathcal{I}} := \mathcal{I}^{\text{Priv}}$ to establish a baseline $\mathcal{I}_{\theta_i}$. We then perform online distillation with $\overline{\mathcal{I}} := \mathcal{I}_{\theta_i}$ to bridge the distribution gap.

## 4.2 SKILL POLICIES

A skill policy $\pi_i$ controls the system for the duration of the skill until the termination condition is satisfied. We train $\pi_i$ using episodic RL. The RL reset distribution is the distribution of final states

following motion planner execution to reach an initiation pose (Sec. 4.1). The 0-1 reward function $r_i(s) := [s \in \mathcal{T}_i]$ is determined by the ground-truth termination condition $\mathcal{T}_i$ (Sec. 4.3).

Although skill learning can be modeled as a standard RL task, the sparse termination condition reward presents exploration challenges, which can even lead to complete learning failure (Stadie et al., 2015; Tang et al., 2017; Ecoffet et al., 2021; Zhou et al., 2024). In light of this, we adopt a residual RL regime (Johannink et al., 2018; Silver et al., 2018; Zhou et al., 2024), starting from a base policy and then fine-tuning it by training an RL agent that outputs differences to the base policy actions, named the residual policy. Specifically, let $\pi^{\text{base}}$ be the base policy and $\pi_{\theta_i}^{\text{res}}$ be the residual policy to learn. We structure our skill policy as: $\pi_{\theta_i}(o_t) = \pi_i^{\text{base}}(o_t) + \pi_{\theta_i}^{\text{res}}(o_t)$. We then train the residual policy with off-the-shelf reinforcement learning algorithms to maximize the regularized objective: $\mathcal{J}(\theta_i) = \mathbb{E}_{\tau_{\text{skill}} \sim \pi_{\theta_i}} \left[ T_i(s_t) - \alpha \cdot \|\pi_{\theta_i}^{\text{Res}}(o_t)\|^2 \right]$. Here, $\alpha$ is a coefficient controlling the regularization strength, which constrains deviations between the base and the fine-tuned policy (Zhou et al., 2024) by penalizing the squared L2 norm of the residual actions; $T_i$ is the termination condition for skill fine-tuning (c.f. Sec 4.3). We choose the skill policy in HSP (Sec. 3.3) as our base policy, which is a behavior-cloning agent trained from the dataset obtained with object-centric data generation.

## 4.3 Termination Classification

A skill termination condition $\mathcal{T}_i$ determines whether the goal of the current stage has been achieved (i.e., current state $s_i \in \mathcal{T}_i$) so that the agent can either enter the next stage or terminate. For deployment, we train a parameterized termination predictor $\mathcal{T}_{\theta_i} : \mathcal{O} \to \{0, 1\}$ from ReinforceGen rollouts with binary cross entropy loss:

$$\mathcal{L}_{\text{term}}(\theta_i) = \mathbb{E}_{\tau_{\text{skill}}}[X_t \log P(\mathcal{T}_{\theta_i}(o_t) = 1) + (1 - X_t) \log P(\mathcal{T}_{\theta_i}(o_t) = 0)], \qquad (3)$$

where $\tau_{\text{skill}}$ is the skill phase trajectory sampled from a trained ReinforceGen agent.

Compared with the initiation pose predictor in Sec. 4.1, an important distinction is that we do not assume limited observability for the termination predictor during training. In principle, we would like to minimize the gap between training and execution. We opt to use ground-truth termination in training to reduce the risk of reward hacking.

**Termination fine-tuning.** In reality, we do not have access to the actual "ground-truth" terminations. Instead, we use hand-crafted conditions with thresholds assigned by experience, which inevitably introduces inaccuracies. Here, we regard the biased termination conditions as a binary predictor that outputs 1 when the current state is deemed a termination and 0 otherwise.

For a binary predictor, two types of errors can occur: false positives, when a termination is predicted but the subtask is not completed, and false negatives, when a subtask is completed but the predictor fails to declare so. False negatives result in a stricter training condition for the RL agent, but ultimately have a limited impact on the performance if the fine-tuned policy can still reach a high success rate. However, false positives can lead to *irrecoverable* states (Fig 3) and can be exploited in RL training.

The termination prediction task can be formulated as an RL problem with a 0-1 action space and the task completion signal as reward. However, this formulation features extremely long horizons and sparse rewards since a decision needs to be made at every timestep. Alternatively, we can focus only on eliminating false positives. Doing so limits the operating space to states where the original predictor determines termination and effectively reduces the horizon due to the sparsity of such states. However, it also largely increases sample collection time, making RL training inefficient. Here, we look at a simplified solution described as follows.

Let $T_i : \mathcal{S} \to \{0, 1\}$ be the termination conditions used for skill fine-tuning. Let $\theta_i^*$ be the parameters fine-tuned with $T_i$. We train a predictor $p_i : \mathcal{S} \to [0, 1]$ to estimate the probability of task success should we terminate the stage at the current state. Formally, $p_i$ is trained to fit $P(s_T \in \mathcal{T} \mid T_i(s_t) = 1, \tau \sim \pi_{\theta_i})$, $\mathcal{T}$ being the task completion conditions. Under the RL formulation of stage terminations, $p_i$ represents the Q-function with $T_i$ as the policy. Let $\mathcal{T}_i^r : \mathcal{S} \to \{0, 1\}$ be the hand-crafted termination conditions and let $T_i$ initialized to $\mathcal{T}_i^r$. Instead of doing RL exploration for an optimal $T_i$, we use a greedy solution by rejecting terminations with a Q-function (i.e., $p_i$) too low. The new termination conditions with a rejection threshold $\epsilon_{\text{term}}$ then is: $T_i(s) := \mathcal{T}_i^r(s) \cdot [p_i(s) > \epsilon_{\text{term}}]$.

## 4.4 REINFORCEGEN HYBRID POLICY

After learning skill initiation conditions, policies, and termination conditions, we can deploy these skills in sequence, using motion planning to connect adjacent pairs of skills. ReinforceGen iterates through each of the $n$ parameterized skills $\psi_{\theta_i}$. For the $i$-th skill, it predicts an end-effector skill initialization pose $p$ from the most recent observation $o$ (Sec. 4.1). Using the motion planner, it plans a trajectory $\tau$ to reach this pose and executes it, while updating the pose target prediction during execution. When the difference between the updated pose and the current motion planner pose exceeds a threshold, ReinforceGen triggers the motion planner to replan to the new pose (Sec. 4.1). Next, it controls the system using the fine-tuned skill policy $\pi_{\theta_i}$ (Sec. 4.4) by predicting and executing actions $a$ until it reaches an observation $o$ that is predicted to correspond to a termination state (Sec. 4.3). We present the pseudocode for this process in Appendix C, Algorithm 1.

## 4.5 END-TO-END DISTILLATION

The hybrid policy in Section 4.4 leverages a motion planner to stich together skill segments. Motion planning requires a model of the world's collision volume, which can be object meshes if the world is fully observable or a point cloud if it not (Sundaralingam et al., 2023). In some deployments, we may wish to bypass the motion planner due to these additional requirements and learn a single end-to-end visuomotor policy. In ReinforceGen, we have the flexibility to do either, where there is a tradeoff between additional requirements and learning difficulty, which a user can tailor to their application.

To train an end-to-end policy, we first configure the motion planner to use the same controller as the skill policy, for example, by switching from joint-space to task-space motion planning control. Then, we can then generate consistent end-to-end trajectories by simply stitching together the motion planning and skill policy segments.

Compared to using baseline imitation agents as the generator, incorporating online data allows ReinforceGen to produce higher-quality source trajectories. Moreover, the random exploration and exploitation process in RL-based fine-tuning makes ReinforceGen agents naturally more resistant to deviations from optimal behaviors. This is especially important in the end-to-end distillation setting, since the deviations compound throughout the long-horizon execution.

## 5 EXPERIMENTS

**Tasks.** We benchmark ReinforceGen on a series of long-horizon manipulation tasks based on robosuite (Zhu et al., 2020). The tasks include the D2 variants (Garrett et al., 2024) (largest initialization range) of the *Nut Assembly*, *Threading*, *Three Piece*, *Coffee*, and *Coffee Preparation* tasks, with up to 5 stages. More details can be found in Appendix B.

**Observation space.** For all tasks, the observation space is comprised of $84 \times 84$ RGB images from a front-view camera and a robot wrist camera as well robot proprioception, namely its end-effector pose and gripper joint position. For the *Threading* task, we include an additional side-view camera since the threading hole can be completely invisible from both the front-view and wrist camera.

**Demonstrations.** We collected 10 human source demonstrations per task. Using ReinforceGen, we automatically adapt these source demonstrations into a BC dataset of 1,000 demonstrations.

**Baselines.** We use HSP-Priv (Sec. 4.1, Garrett et al. (2024)) as our baseline with privileged information. We further fine-tune its skill policies to establish an upper-bound on performance (HSP-Priv + Skill-FT). For baselines sharing the same assumptions, we use our implementation of HSP - distilled with online rollouts with HSP-Priv as the teacher policy.

**Evaluation setup.** We partially disable fine-tuning on stages that already have high baseline success rates to save computational cost. For an exhaust implementation list, refer to App. C.5. We use hand-crafted stage terminations in our evaluations for all hybrid policies despite it requiring state information, which is in line with prior works (Garrett et al., 2024; Zhou et al., 2024). We instead include an ablation in Sec. 5.2 that uses learned terminations.

| Success Rate (%) | Nut Assem. | Threading | Three Piece | Coffee | Coffee Prep. | Overall |
|---|---|---|---|---|---|---|
| SPIRE (Zhou et al.) | N/A | N/A | 86.00 | 98.00 | 84.00 | N/A |
| HSP-Priv | 86.20 | 54.27 | 70.77 | 88.12 | 65.60 | 72.99 |
| HSP-Priv + Skill-FT | 87.20 | 89.40 | 82.24 | 97.03 | 77.80 | 86.66 |
| HSP | 40.52 | 50.20 | 41.52 | 55.38 | 35.80 | 44.68 |
| HSP + Replan | 78.40 | 49.80 | 65.80 | 80.20 | 60.20 | 66.88 |
| ReinforceGen (Ours) | **85.80** | **82.20** | **80.40** | **93.81** | **80.80** | **84.60** |

**Table 1:** Comparing task success rate across all tasks. SPIRE uses 200 demonstrations while the rest use 10. SPIRE, HSP-Priv (Sec. 4.1), and HSP-Priv + Skill-FT use privileged state information, while the rest rely only on observations. The numbers are averaged from over 500 rollouts except for SPIRE, which uses 50.

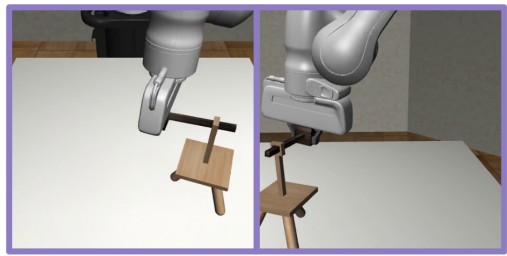 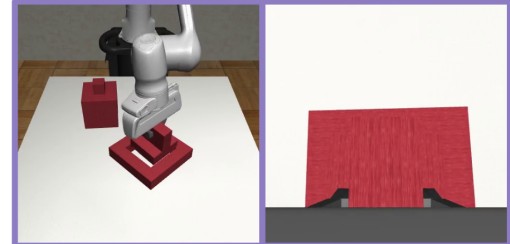

**(a)** *Threading* (Stage 2) - **82.2%** Success Rate          **(b)** *Three Piece* (Stage 2) - **93.8%** Success Rate

**Figure 5:** ReinforceGen agents complete high-precision skills with high success rates.

## 5.1 MAIN RESULTS

**ReinforceGen is proficient in all tasks despite partial observability.** Tab. 1 shows that Reinforce-Gen achieves over **80%** task completion rate in all our tested tasks, including long-horizon tasks with as many as 5 stages, all while relying only on camera sensory input and proprioception information.

**ReinforceGen achieves 89% relative performance increase to baselines without state observability on average.** In Tab. 1, comparing HSP-Priv and HSP, we observe a significant drop in success rates due to the lack of object location awareness. With the same accessible information, ReinforceGen achieves almost double the performance against HSP in overall success rate. We also notice that our advantages are especially higher in longer-horizon tasks with 4-5 stages, i.e., *Nut Assembly*, *Three Piece*, *Coffee Preparation*, reaching **109%** relative increase.

**Even compared with baselines with privileged information, ReinforceGen is still competitive.** We fine-tuned HSP-Priv with access to state information as a pseudo-performance upper bound (HSP-Priv + Skill-FT in Tab. 1). Despite the disadvantages, ReinforceGen only shows a maximum deficit of **8%** in *Threading*, with **2%** decrease in the overall success rate.

**ReinforceGen agents reliably complete tasks with a small margin of error.** As shown in Fig. 5, our benchmark task set involves multiple high-precision skills that are challenging to perform, especially when exact object poses are not observed. For ReinforceGen, continuously updating target prediction with the latest observations (Sec. 4.1) reduces the randomness in the skill policy starting conditions, and our RL-based skill fine-tuning (Sec. 4.4) explores and improves the suboptimal imitation skills.

## 5.2 ABLATION RESULTS

**Pose target replanning improves motion planning reaching accuracy and skill completion.** Comparing HSP and HSP-Replan in Table 1, adding replanning alone increases the overall success rate of HSP by **22%** (from 44.68% to 66.88%), a relative **50%** improvement. We present a more detailed case study in *Nut Assembly* in Appendix D.1.

**Skill policy fine-tuning recovers inaccurate starting poses and suboptimal imitated skills.** We ablate skill fine-tuning from ReinforceGen to show its effectiveness in Tab. 2. On average, skill fine-tuning improves task completion rates by **24.41%** while using **9.74%** less steps to complete. Skill fine-tuning is most effective in the second stage of *Threading*, which is the most precision-demanding stage among all benchmarked tasks, and where the IL agent has the lowest success rate.

| Improvement (%) | Nut Assembly | Threading | Three Piece | Coffee |
|---|---|---|---|---|
| Success Rate | 4.62 | 65.73 | 10.55 | 16.74 |
| Efficiency | 8.43 | 16.39 | 10.18 | 3.97 |

**Table 2:** ReinforceGen's policy improvement through skill fine-tuning. Efficiency is the average number of timesteps to complete each skill. Success Rate and Efficiency are averaged over all task stages.

**Termination fine-tuning repairs cross-stage causal effects.** Termination fine-tuning bridges independently-learned stages. For example, in the *Three Piece* task, ReinforceGen recognizes a failure mode where placing the first piece on the edge of the base frame causes the placement of the second piece to fail. When directly applied to HSP-Priv, termination fine-tuning improves its success rate from 66.44% to 70.77%; After fine-tuning skills with the new terminations, the success rate further increases from 72.60% to 82.24%.

**Learned termination predictor has limited impact on task completion.** Finally, we evaluate ReinforceGen with a learned termination predictor that determines termination with partial observations. As shown in Table 3, using a learned termination predictor only introduces minor drops in success rates in *Threading* and *Three Piece*, while maintaining the performances in the rest.

| Success Rate (%) | Nut Assembly | Threading | Three Piece | Coffee |
|---|---|---|---|---|
| Oracle Termination | 85.80 | 82.20 | 80.40 | 93.81 |
| Learned Termination | 84.60 | 79.20 | 73.80 | 92.60 |

**Table 3:** ReinforceGen's learned termination predictor, which unlike the Oracle Termination does not have access to the state, performs well albeit slightly worse when compared to the Oracle Termination upper bound.

## 5.3 END-TO-END RESULTS

Finally, we attempt to distill the hybrid ReinforceGen agents to end-to-end visuomotor policies. To do so, we first configure the motion planner's action space to be consistent with the skill policy, allowing us to stitch the connect and skill phase trajectories of each stage together. We then create a dataset of such rollouts for each environment and train a BC agent from them. The result is shown in Table 4.

In *Threading* and *Coffee*, ReinforceGen distills to proficient end-to-end agents, with significant improvement over the baseline. However, in *Nut Assembly* and *Three Piece*, both agents struggle to complete the task. We observe that these two tasks feature long-range transfers with significant rotational movements. As a result, the distilled agent often ends up in joint lock-up states in the connect phases. This is one of the limitations of our work that we hope to improve in future works.

| Success Rate (%) | Nut Assembly | Threading | Three Piece | Coffee |
|---|---|---|---|---|
| HSP-Priv | **35.00** | 60.40 | **20.20** | 84.60 |
| ReinforceGen-Priv | 28.00 | **83.60** | 18.60 | **94.20** |

**Table 4:** End-to-end distillation results using datasets generated with ReinforceGen-Priv and HSP-Priv.

## 6 LIMITATIONS

Our demonstration system is not fully automated as we still assume access to a handful of human demonstrations (e.g. 1 - 10). In order to adapt skill demonstrations, we assume access to reference frame labeled per skill, which might be difficult to specify for skills that don't involve conventional objects, for example a sweep skill involving granual media. In our experience, we train specialized task-conditioned policies. Future work involves investigating the extent to which skill policies can be used across tasks. We use the observed point cloud as the collision volume for motion planning, which might be the insufficient in situations with heavy partial observability.

## 7 CONCLUSION

We presented ReinforceGen, an automated demonstration generation system that integrates planning, demonstration adaptation, and reinforcement learning to bootstrap a small set of human demonstrations into a large high-quality dataset. At deployment time, these demonstrations can be used to train

end-to-end imitation learners or atomic skill policies that are stitched together using motion planning, demonstrating the flexibility of our approach. We showed that the use of reinforcement learning for exploration ultimately improves policy success rates when compared to prior works due to its ability to explore beyond adapted demonstrations.

## 8    REPRODUCIBILITY STATEMENT

We provide the details to reproduce our results in Appendix C.

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

## A OVERVIEW

- Appendix B shows the details of the benchmark task set.
- Appendix C lists the details to reproduce our results.
- Appendix D includes additional experiment results.
- Appendix E includes additional implementation details.

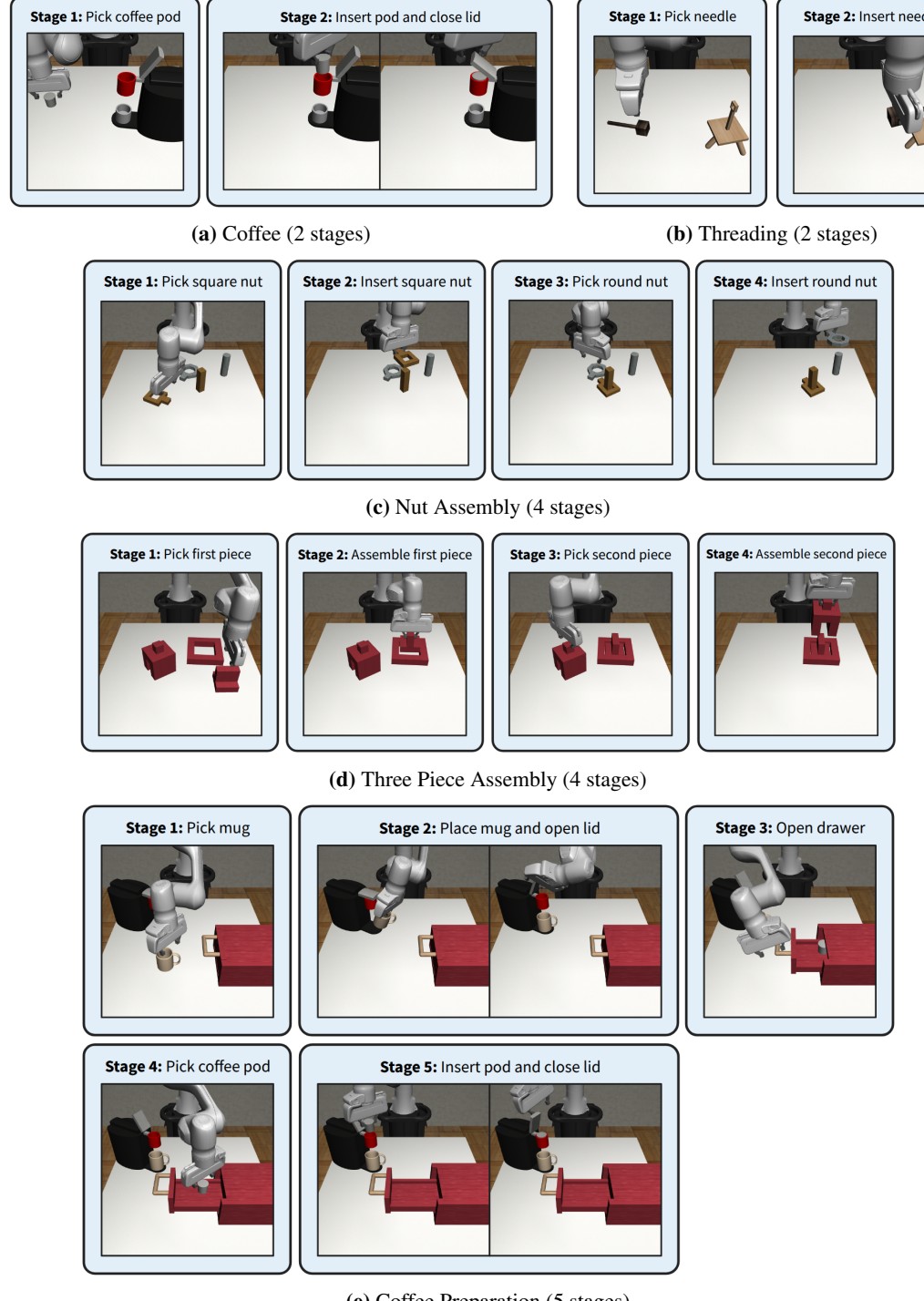

**(a)** Coffee (2 stages)  **(b)** Threading (2 stages)

**(c)** Nut Assembly (4 stages)

**(d)** Three Piece Assembly (4 stages)

**(e)** Coffee Preparation (5 stages)

**Figure 6:** All five tasks showcased.

## B TASKS

We choose the five tasks from Robosuite (Zhu et al., 2020; Garrett et al., 2024) (Fig. 6). We use the largest initiation range version (D2) for all tasks. The only exception is *Nut Assembly*, where we reduce the $x$-range of the nuts placement to $(-0.15, 0.15)$ since the original range produces unreachable initial positions.

## B.1 OBSERVATIONS

The observation consists of two (or three, in Threading) 84×84 RGB images and robot proprioception state, including a 6-dim end-effector pose composed of 3-dim Euclidean position in meters and 3-dim rotation represented in axis-angle form; a 2-dim gripper position, representing the distances of the two grippers to the center; and a 7-dimensional joint configuration. The front-view camera is demonstrated in Figure 6; the wrist camera angle is shown in Figure 5b (right); the extra Threading side-view camera is shown in Figure 5a (right).

# C  REPRODUCIBILITY

## C.1  PSEUDOCODE

---

**Algorithm 1** ReinforceGen Deployment Pseudocode

---

1:  **procedure** REINFORCEGEN
2:      $o \leftarrow$ `env.reset()`                                          ▷ Get initial observation
3:      **for** $i := 1 \rightarrow n$ **do**
4:          $\langle R_i, \mathcal{I}_{\theta_i}, \pi_{\theta_i}, \mathcal{T}_{\theta_i} \rangle \leftarrow \psi_{\theta_i}$                ▷ Skill $\psi_i$
5:          $p \leftarrow \mathcal{I}_{\theta_i}(o)$                                        ▷ Predict initiation pose
6:          $\tau \leftarrow$ `planToPose(env, p)`                            ▷ Motion planning
7:          **for** $a \in \tau$ **do**
8:              $o \leftarrow$ `env.step(a)`                                  ▷ Execute motion action
9:              $p' \leftarrow \mathcal{I}_{\theta_i}(o)$
10:             **if** $\text{dis}(p, p') > \epsilon$ **then**                          ▷ Refined initiation prediction
11:                 $p \leftarrow p'$
12:                 $\tau \leftarrow$ `planToPose(env, p)`                    ▷ Replan trajectory
13:         **while** $\mathcal{T}_{\theta_i}(o) \neq 1$ **do**                              ▷ Until subgoal success
14:             $a \leftarrow \pi_{\theta_i}(o)$
15:             $o \leftarrow$ `env.step(a)`                                  ▷ Execute policy action

---

## C.2  THRESHOLDS USED IN REINFORCEGEN

**Threshold for motion planner replanning (Sec. 4.1)** We replan the motion planning trajectory when the pose distance (c.f. App. E.2) between the newest predicted pose and the current pose target exceeds **0.05**.

**Threshold for termination rejection (Sec. 4.3)** We reject terminations with a predicted task completion rate lower than **0.4**.

## C.3  HYPERPARAMETERS FOR DATA GENERATION AND IMITATION LEARNING

We follow the exact setup in Garrett et al. (2024).

## C.4  HYPERPARAMETERS FOR REINFORCEMENT LEARNING

**Table 5:** DrQ-v2 hyperparameters.

| | |
|---|---|
| Network structure | CNN |
| Learning rate | 1e-4 |
| Discount | 0.99 |
| Batch size | 256 |
| $n$-step returns | 3 |
| Action repeat | 1 |
| Seed frames | 4000 |
| Feature dim | 50 |
| Hidden dim | 1024 |
| Optimizer | Adam |
| Training steps | 2M |
| Constraint coef. ($\alpha$) | 5.0 |

We use DrQ-v2 (Yarats et al., 2021) for skill fine-tuning (Sec. 4.4). The hyperparamters are shown in Tab. 5.

## C.5  USAGE OF FINE-TUNING METHODS

By default, all stages in all tasks use first-iteration pose distillation and real-time replanning (Sec. 4.1). The usage of second-iteration pose distillation and skill/termination fine-tuning is listed in Tab. 6.

**Table 6:** Usage of different fine-tuning methods in ReinforceGen

| Task | Stage | Pose distillation (Sec. 4.1) | Skill fine-tune (Sec. 4.4) | Termination fine-tune (Sec. 4.3) |
|------|-------|:---:|:---:|:---:|
| Coffee | 1 | ✗ | ✗ | ✗ |
|  | 2 | ✗ | ✓ | ✗ |
| Threading | 1 | ✗ | ✗ | ✗ |
|  | 2 | ✗ | ✓ | ✗ |
| Nut Assembly | 1 | ✓ | ✗ | ✗ |
|  | 2 | ✗ | ✓ | ✗ |
|  | 3 | ✓ | ✓ | ✗ |
|  | 4 | ✗ | ✗ | ✗ |
| Three Piece | 1 | ✗ | ✗ | ✓ |
|  | 2 | ✗ | ✓ | ✓ |
|  | 3 | ✗ | ✗ | ✓ |
|  | 4 | ✗ | ✓ | ✓ |
| Coffee Prep. | 1 | ✗ | ✗ | ✗ |
|  | 2 | ✗ | ✓ | ✗ |
|  | 3 | ✗ | ✓ | ✗ |
|  | 4 | ✗ | ✓ | ✗ |
|  | 5 | ✗ | ✓ | ✗ |

## C.6 END-TO-END DISTILLATION

We add Gaussian noise $\mathcal{N}(0, 1)$ with $\sigma = 0.01$ during trajectory generation. For each task, we generate 3000 successful trajectories. We use the same hyperparameters as skill imitation for IL.

# D    ADDITIONAL RESULTS

## D.1    CASE STUDY ON THE EFFECTIVENESS OF REPLANNING IN *Nut Assembly*

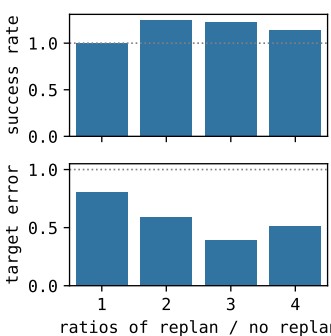

**Figure 7:** Ablation of real-time replanning on every stages of *Nut Assembly*. The top figure shows the per-stage success rate, and the bottom shows the pose target error. The numbers are ratios of applying replanning versus not applying.

To illustrate the effectiveness of real-time replanning (Sec. 4.1), we plot the reduction in prediction error and improvement in task success rate with replan in all 4 stages in *NutAssembly*. The results in Fig. 7 show that replan significantly reduces the target prediction error in all stages, and in turn, improves the subsequent skill success rates.

## D.2    ABLATION ON DISTILLATION

In Fig. 8, we add artificial noises to the actions and plot the relative performance decreases against the noise scale. In all tasks, ReinforceGen appears to be more resistant to action noises and trajectory deviations, matching our assertions in Sec. 4.5.

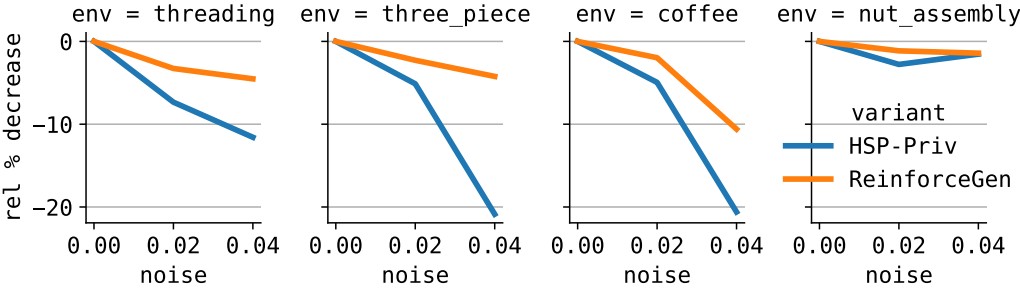

**Figure 8:** Comparing the resistance to action noise between HSP-Priv and ReinforceGen. For most tasks, ReinforceGen has a much higher tolerance to action noise.

# E ADDITIONAL DETAILS

## E.1 POSE NOISE IN SECTION 4.1

In Figure 4, we add artificial noise to initiation poses to demonstrate their relationship with skill completion rates. The noises are added with the following procedure. Let the original pose be $\boldsymbol{P} := \boldsymbol{p} \oplus \boldsymbol{r}$, where $\boldsymbol{p}$ is a 3-dimensional position vector in meters, $\boldsymbol{r}$ is a 3-dimensional rotation vector in the axis-angle form. A noised version of $\boldsymbol{P}$ with a scale $\sigma$ is defined as:

$$\tilde{\boldsymbol{P}} \leftarrow \boldsymbol{P} + \mathcal{N}(\boldsymbol{0}, \sigma^2 \boldsymbol{I}_6). \tag{4}$$

## E.2 POSE DISTANCE METRIC

We use the following metric to compute the distance between two poses throughout our implementation. Let $\mathbf{P_1} := (\mathbf{p_1}, \mathbf{q_1})$, $\mathbf{P_2} := (\mathbf{p_2}, \mathbf{q_2})$ be the two poses to compare, $\mathbf{p_1}, \mathbf{p_2}$ the Euclidean positions, and $\mathbf{q_1}, \mathbf{q_2}$ the quaternion-form rotations.

$$d^{\mathrm{pos}}(\mathbf{P_1}, \mathbf{P_2}) := \sqrt{(\mathbf{p_1} - \mathbf{p_2})^\top (\mathbf{p_1} - \mathbf{p_2})} \tag{5}$$

$$d^{\mathrm{rot}}(\mathbf{P_1}, \mathbf{P_2}) := \frac{1}{\pi} \min\{\cos^{-1}(\mathbf{q_1}^\top \mathbf{q_2}), \cos^{-1}(-\mathbf{q_1}^\top \mathbf{q_2})\} \tag{6}$$

$$d^{\mathrm{pose}}(\mathbf{P_1}, \mathbf{P_2}) := d^{\mathrm{pos}}(\mathbf{P_1}, \mathbf{P_2}) + d^{\mathrm{rot}}(\mathbf{P_1}, \mathbf{P_2}) \tag{7}$$

