# OpenReview forum: "ReinforceGen: Hybrid Skill Policies with Automated Data Generation and Reinforcement Learning"
_ICLR.cc/2026/Conference — Submitted to ICLR 2026_

### Official Review · Reviewer_1Aqh · 2025-10-21

**Soundness:** 3
**Presentation:** 2
**Contribution:** 2
**Rating:** 4
**Confidence:** 3

**Summary:**

This work extends an object-centric data-generation pipeline, SkillMimicGen, to long-horizon manipulation, introducing ReinforceGen, a hybrid approach that alternates between motion planning and skill execution while fine-tuning three components comprising the pose predictor, the skill policy, and the termination predictor. Improve the skill policy with residual RL on top of a BC prior; use online replanning at initiation and teacher distillation during training for better generalization; and reduce termination false positives by including a success-probability gate. Results demonstrate that with limited human demonstrations, this approach outperforms prior SkillMimicGen-style baselines on five Robosuite tasks and is comparable in performance to policies using privileged state.

**Strengths:**

* Hybrid pipeline using "BC + RL": explicit decomposition with clear initiation/skill/termination, targeted improvements using replanning, residual RL, termination gating.
* Competitive performance with only a few human demos; significantly outperform prior SkillMimicGen-based baselines and close to privileged-state “upper bounds”.
* Demonstrate the possibility of incorporating online RL into a generation pipeline centered on objects with no discarding of planning advantages.

**Weaknesses:**

* Narrow scope of evaluation: Only five tasks in Robosuite; greater task diversity, and another simulator, would strengthen external validity.
* No real-robot validation: Without any hardware result, its applicability remains uncertain under real world settings.
* Presentation clarity: Many implementation details (e.g., thresholds, reset distributions, termination gate calibration) are somewhat opaquely documented.

**Questions:**

* Hyperparameter sensitivity: How sensitive is performance to planning replan threshold for initiation and termination gate threshold? Learning curves or sensitivity plots across reasonable range for the hyperparameters may provide clarity.
* Algorithm selection: Why DrQ-v2? Would PPO, SAC, IQL yield similar gains under the residual setup and image inputs? Any ablations across RL algorithms?
* Embodiment Generality: Is there an extension to other embodiments, for example, multi-finger hands/dexterous manipulation? What will be the expected pain points?

---

> ### Author Response · Authors · 2025-11-26
>
> We thank the reviewer for their constructive comments and feedback. Below, we addressed the reviewer’s concerns and questions.
>
> *"Narrow scope of evaluation"*
> - Robosuite is a widely used benchmark for multi-stage manipulation tasks. The five tasks that we benchmarked cover grasping objects of different shapes, precision insertion, multi-step assembly, and articulated object manipulation. We believe this set of tasks is representative of common robotic manipulation tasks, as recognized in multiple prior works [1, 2, 3]. By design, our method is applicable to different benchmarks/simulators if the simulator offers access to object state information in training and a black-box motion planner.
>
> *"No real-robot validation"*
> - We acknowledge the lack of real-world experiments. In light of this, we propose a sim-to-real transfer procedure for ReinforceGen and argue that simulation-only experiments are also valuable. We kindly refer the reviewer to our common response for a detailed discussion.
>
> *Missing implementation details*
> - Thresholds
>   - Pose difference threshold for initial pose replanning (Section 4.1): **0.05**
>   - Rejection threshold for termination fine-tuning (Section 4.3): **0.4**
>   - We have now included the above information with more detail in Appendix C.2 in our updated manuscript.
> - Reset distributions
>   - We use the D2 reset distribution from the Robosuite task set used in SkillGen [2]. The only exception is Nut Assembly, where we reduce the x-range of the nuts placement to $(-0.15, 0.15)$ since the original range produces unreachable initial positions. The above information is presented in Appendix B.
>
> *"Why DrQ-v2?"*
> - We follow prior work [3] for the choice of the RL algorithm. Other algorithms may reach similar performances to DrQ-v2, but we consider this to be orthogonal to our work. The design of ReinforceGen allows users to select any off-the-shelf RL algorithms for different tasks.
>
> *Embodiment Generality*
> - To generalize to dexterous manipulation, we can follow the practice in [4]. Specifically, during data generation, we can replay the finger-joint actions in the source demo, as the finger movement is always relative to the end-effector. During residual RL finetuning, we can append the finger-joint actions to the original pose command of the end-effector and regularize them accordingly. The increase in action dimension may bring additional challenges to exploration.

---

> > ### Comment · Reviewer_1Aqh · 2025-11-27
> >
> > Thank you for the additional details and the concise rebuttal.
> > My main concern remains the lack of real-world evaluation. Similar works, such as MimicGen and DexMimicGen, include real-robot experiments and extensive analysis. In contrast, the current paper proposes a theoretically feasible pipeline in rebuttal, but in robotics, demonstrations on real hardware are extremely important.
> > I will keep my original score. I hope the authors can understand the necessity of real-world experiments. There are too many tricks in simulators that can make a method appear to work.

---

> ### Author Response · Authors · 2025-11-26
>
> ## References
>
> [1] Mandlekar et al., Human-in-the-Loop Task and Motion Planning for Imitation Learning, 2023
>
> [2] Garrett et al., Skillmimicgen: Automated demonstration generation for efficient skill learning and deployment, 2024
>
> [3] Zhou et al., SPIRE: Synergistic Planning, Imitation, and Reinforcement Learning for Long-Horizon Manipulation, 2024
>
> [4] Jiang et al., DexMimicGen: Automated Data Generation for Bimanual Dexterous Manipulation via Imitation Learning, 2024

---

### Official Review · Reviewer_GL2P · 2025-10-28

**Soundness:** 2
**Presentation:** 2
**Contribution:** 1
**Rating:** 2
**Confidence:** 4

**Summary:**

The work proposes a pipeline of methods to learn a policy for robotic tasks from offline data and then fine-tune it with several expert demonstrations. In limited experimental setup, the method seems to improve over a baseline.

**Strengths:**

The method addresses a real problem in robot learning - learn to solve a task from a substantially small number of expert demonstrations. Learning is divided to offline and online parts. This makes sense since the compounding error can be reduced by the online part (in the same spirit as Dagger).

**Weaknesses:**

**Major:**

 - The work claims to contribute to robot learning, but no experiments with real robots are made - everything is simulated

 - The work is a good piece of engineering work where certain parts from existing works are put together and then experimented, but lacks  in scientific research questions or contributions

 - Results table does not include comparison to existing works despite that for the used tasks related works have been published




**Moderate:**

 - You claim that during training you have access to the object state (pose), but this is straightforward only for simulated environments - how about real environments, how this is realized without substantial extra equipment & calibration etc.?

 - In Figure 3 and text - why you use the term "distillation" as I assume you just train a pose predictor?

 - In Section 4.2 you list methods and then plainly justify that the *residual RL* is selected. I assume any recent offline RL methods should work here and if not, then some kind of comparison should be provided.

 - Clearly define what are the observable inputs to the system during inference

**Minor:**

 - what is the unit in Figure 4 since pose consists of three metric translation component and three angular orientation components? I don't think the success rate drops "sharply" - this is especially difficult to judge without knowing the units.

 - Section 4.4 assumes that a reader reads Appendix C at the same time - without the appendix the section is useless. This is not good practice as every article should be sufficiently self-contained

**Questions:**

This work is a good piece of engineering work, but to make it stronger and more scientific:

 - Compare to known SotA on these benchmarks and in fair settings (should be easy as everything happens in simulations)
 - IF your approach is clearly better than SotA explain what parts and why they solve the problem/limitation in the existing methods
 - Demonstrate your results with real robot tasks (simulation is not robotics as the real world brings in many challenges)

---

> ### Author Response · Authors · 2025-11-26
>
> We thank the reviewer for their constructive comments and feedback. Below, we addressed the reviewer’s concerns and questions.
>
> *"... no experiments with real robots are made."*
> - We acknowledge that our experiments are only in simulation. In light of this, we propose a sim-to-real transfer procedure for ReinforceGen and argue that simulation-only experiments are also valuable. We kindly refer the reviewer to our common response for a detailed discussion.
>
> *"... lacks in scientific research questions or contributions."*
> - We appreciate the reviewer recognizing our work as “a good piece of engineering work”. We’d like to point out that ReinforceGen is also a novel algorithm that outperforms previous works, and we did study important research questions and have scientific research contributions.
>   - A successful integration of different components requires structured and iterative exploration, which constitutes scientific research.
>   - Several prior works [1, 2, 3] in a similar setting have conducted their research similarly. Our research improves upon those works and thereby offers meaningful scientific contributions.
>   - We studied important research questions in our paper. In Section 5.1, we put our performance into context by comparing ReinforceGen with both baselines under the same assumptions (HSP) to demonstrate the improvement, and baselines with stronger assumptions (SPIRE, HSP-Priv), showing that our achieved result is close to the upper bound. In Section 5.2 and Appendix D, we ablate the different components of ReinforceGen to analyze their individual contributions.
>
> *"... does not include comparison to existing works, ..."*
> - We compare ReinforceGen to two recent works, SkillGen [2], which shares most our assumptions, and SPIRE [3], which is a privileged baseline with stronger assumptions to establish an upper bound. Both works are strong baselines published within one year of this submission. If the reviewer has other suggestions, we would be happy to incorporate them into our comparisons.
>
> *Access to the object state in real world training:*
> - For initial data generation, we can follow the setup in SkillGen ([2], Section 6.3 and Appendix K) that uses a combination of RealSense cameras and FoundationPose [4] for object pose estimation. In most of our fine-tuning process, we only require imagery and proprioceptional inputs. The two occasions that requires object state access are:
>   1. Training a student pose predictor from a privileged teacher. In this case, the teacher predictor only requires access to the pose of a static object once in the beginning of the motion planning phase. This is similar to the SkillGen data generation setting. Therefore we conjecture that the same setup can also work here.
>   2. Training the RL agent uses a termination classifier with object state information. We can still follow the same setup in SkillGen for pose estimation and use termination fine-tuning (Section 4.3) to resolve the potential inaccuracies.
> - Alternatively, we can use a sim-to-real pipeline, leaving training completely in simulation. We describe this pipeline in more detail in our general response.
>
> *"Why you use the term 'distillation'?"*
> - We apologize for the inaccurate wording. We have updated our manuscript accordingly. In this context, we are training a student pose predictor from a teacher predictor with privileged object state access.
>
> *"... any recent offline RL methods should work here, ..."*
> - ReinforceGen focuses on integrating online interaction data to improve the quality of data generation. Therefore, offline RL, which only uses offline data during training, is not a good fit here. Residual RL is a commonly used method in online RL with offline data regime, which suits our objective.
>
> *"Clearly define what are the observable inputs to the system during inference."*
> - The observation space is stated in Section 5 L357-360. The observation consists of two (or three, in *Threading*) 84x84 RGB images and robot proprioception state, including a 6-dim end-effector pose composed of 3-dimensional Euclidean position in meters and 3-dimensional rotation represented in axis-angle form; a 2-dimensional gripper position, representing the distances of the two grippers to the center; and a 7-dimensional joint configuration. The front-view camera is demonstrated in Figure 6 in Appendix B; the wrist camera angle is shown in Figure 5b (right); the Threading side-view camera is shown in Figure 5a (right). We have added a new section in Appendix B to describe the observation in detail.

---

> ### Author Response · Authors · 2025-11-26
>
> *"What is the unit in Figure 4"*
> - We added the details of adding noises to predicted poses in Appendix E.1 and a reference to it under the caption of Figure 4. In short, we represent poses as a concatenation of a 3-dimensional position vector in meters and a 3-dimensional rotation vector in axis-angle form, and add Gaussian noise with corresponding scales.
>
> *"Section 4.4 assumes that a reader reads Appendix C"*
> - We have updated Section 4.4 to make it self-contained.
>
> ## References
>
> [1] Mandlekar et al., Human-in-the-Loop Task and Motion Planning for Imitation Learning, 2023
>
> [2] Garrett et al., Skillmimicgen: Automated demonstration generation for efficient skill learning and deployment, 2024
>
> [3] Zhou et al., SPIRE: Synergistic Planning, Imitation, and Reinforcement Learning for Long-Horizon Manipulation, 2024
>
> [4] Wen et al., FoundationPose: Unified 6D Pose Estimation and Tracking of Novel Objects, 2023

---

### Official Review · Reviewer_7A7r · 2025-11-01

**Soundness:** 2
**Presentation:** 2
**Contribution:** 2
**Rating:** 4
**Confidence:** 4

**Summary:**

Building on the Hybrid Skill Policy (HSP) paradigm, the paper decomposes tasks into multiple sub-tasks. It then proposes a demonstration/data-generation augmentation pipeline that couples motion planning with reinforcement learning: starting from a small number of human demonstrations, the method expands to a larger offline dataset and further refines the policy via RL. On five long-horizon robosuite tasks, it outperforms a vanilla HSP baseline, though it still lags behind approaches trained with substantially more human demonstrations and those leveraging privileged state information.

**Strengths:**

1. Termination Classification: The authors introduce a learned termination classifier that minimizes the gap between training and deployment by rejecting low-confidence terminations, thereby reducing train–test mismatch during execution.

2. Initiation Pose Prediction: The initiation pose predictor is continuously updated during the connection segment and can trigger replanning when necessary, which substantially reduces pose error and improves task success rates.

**Weaknesses:**

1. Rationale for Imitation Learning vs. Pure RL: It is unclear why imitation learning (IL) is necessary. How would a purely reinforcement-learning pipeline perform under the same training budget and environment settings? Does IL primarily improve computational efficiency (sample/compute efficiency) or final success rate—and by how much?

2. Gap to Stronger Oracles and Data Regimes: Although the method improves over a vanilla HSP baseline, it still underperforms settings that use privileged state information and/or substantially larger human-demo corpora.

3. Termination Classifier Reporting: While a termination classification scheme is proposed, the paper does not report its standalone accuracy quality.

4. Lack real-world setting to verify the capability of model.

**Questions:**

Reinforcement learning appears to contribute substantially to performance. How do you quantify RL’s contribution?

---

> ### Author Response · Authors · 2025-11-26
>
> We thank the reviewer for their constructive comments and feedback. Below, we addressed the reviewer’s concerns and questions.
>
> *"Imitation Learning vs. Pure RL"*
> - Pure online RL is known to struggle in continuous control tasks [1, 2], especially with sparse reward signals [3]. SPIRE [3], which shares a similar benchmark set as ours with much stronger assumptions, shows that pure RL struggles to learn anything beyond the easiest task. Their work set the upper-bound performance for pure RL in our setting, which is why we excluded pure RL experiments.
>
> *"Gap to Stronger Oracles and Data Regimes"*
> - **Gap to stronger oracles**: In Table 1, compared with the privileged baseline SPIRE, our method generally underperforms by less than 6%. Compared with HSP-Priv + Skill-FT, a baseline that is closer to our assumption but still uses privileged state information, we only underperform by 2% overall. We consider this a reasonable gap. The goal of our work is to improve baseline performances with fewer assumptions, which we have demonstrated to achieve.
> - **Gap to higher data regimes**: The claim is that ReinforceGen could underperform baselines with substantially more human demonstrations. While we cannot exclude the possibility, we want to point out that demo-efficiency is one of the major advantages of ReinforceGen. For each of the benchmarked tasks, we only use 10 source human demonstrations to reach the reported performance.
>
> *"Lack real-world setting"*
> - We acknowledge the lack of real-world experiments. In light of this, we propose a sim-to-real transfer procedure for ReinforceGen and argue that simulation-only experiments are also valuable. We kindly refer the reviewer to our common response for a detailed discussion.
>
> *"Quantify RL’s contribution"*
> - The RL’s contribution is quantified in our ablation study on skill fine-tuning (Table 2, Section 5.2, paragraph 2). The relative improvement in task completion rate from RL-based skill fine-tuning is 24.41%, averaged over benchmarked tasks.

---

> > ### Comment · Reviewer_7A7r · 2025-11-27
> >
> > Unfortunately, the authors did not address the main issues I raised, so I will maintain my current score.

---

> > > ### Author Response · Authors · 2025-11-27
> > >
> > > Please specify the main issues you feel were not sufficiently addressed so we can improve our response.

---

> ### Author Response · Authors · 2025-11-26
>
> ## References
>
> [1] Johannink et al., Residual Reinforcement Learning for Robot Control, 2018
>
> [2] Ball et al., Efficient Online Reinforcement Learning with Offline Data, 2023
>
> [3] Zhou et al., SPIRE: Synergistic Planning, Imitation, and Reinforcement Learning for Long-Horizon Manipulation, 2024

---

### Official Review · Reviewer_NVAj · 2025-11-01

**Soundness:** 2
**Presentation:** 2
**Contribution:** 2
**Rating:** 4
**Confidence:** 3

**Summary:**

The paper introduces a learning pipeline for multi-stage robotic manipulation tasks, including a MimicGen and imitation learning stage followed by an online reinforcement learning (RL) stage. During the online RL stage, initial pose predictors and termination classifiers are learned for each subtask for subtask stitching and success detection, and residual RL is used to improve each subtask policy. Finally, all components are distilled into  an end-to-end policy for vision-based execution without privileged information.
Experiments on some simulation tasks demonstrate that the system outperforms HSP baselines and achieves comparable performance against SPIRE trained on more human demonstrations.

**Strengths:**

- RL finetuning is a significant problem for MimicGen-style robotic manipulation policies.
- The system dividing the long-horizon task into motion-planning stages and RL stages effectively reduces the burdens for RL exploration.
- Experimental results demonstrate its effectiveness compared with the previous HSP method.

**Weaknesses:**

- The RL pipeline only works in simulation, since privileged states are required to train the initial pose predictors and termination classifiers. Given this, is it really necessary to train these modules? A much simpler approach may also works: using ground truth motion-planning poses and ground truth success detectors for RL in simulation, then distilling all modules into an end-to-end policy.
- Another simple and direct approach is not compared with: apply MimicGen, BC, and residual RL for each individual subtask in simulation, then alternately execute motion planning and these policies for distillation.
- Lacks discussion and comparison of previous works solving the similar problem [1,2,3].
- The sim-to-real performance and real-world applicability of the system is not evaluated.

[1] Chen et al., Sequential Dexterity: Chaining Dexterous Policies for Long-Horizon Manipulation, 2023
[2] Agia et al., STAP: Sequencing Task-Agnostic Policies, 2023
[3] Lee et al., Adversarial Skill Chaining for Long-Horizon Robot Manipulation via Terminal State Regularization, 2021

**Questions:**

- Please see weaknesses.
- Will synthesizing more data (e.g. 10000 demos) during the MimicGen stage bring the same improvement?
- Why is DrQ-v2 used for residual RL? Do other popular RL algorithms (e.g. PPO and SAC) work?

---

> ### Author Response · Authors · 2025-11-26
>
> We thank the reviewer for their constructive comments and feedback. Below, we addressed the reviewer’s specific concerns and questions.
>
> *“... using ground truth motion-planning poses and ground truth success detectors for RL in simulation, then distilling all modules into an end-to-end policy.”*
> - We agree with the reviewer that directly distilling ground-truth motion planning trajectories and privileged RL policies can potentially be a strong baseline. However:
>   - ReinforceGen shares the same observation space as the final policy and thus can learn behaviors to compensate for the lack of privileged information. ReinforceGen incorporates corrective mechanisms both explicitly from real-time replanning (Section 4.1) and implicitly from RL-based skill finetuning (Section 4.2) to handle observation limitations. Such corrective behavior can potentially benefit the distilled policies with the same limitations.
>   - “Ground-truth” success detectors are still designed by humans and can be defective and exploited by RL. An example in Figure 3C shows that an inaccurate piece placement is deemed a  success by the “ground-truth” success detector due to imperfect thresholding. Our termination fine-tuning process (Section 4.3) helps alleviate this issue. This is empirically supported in Section 5.2, “Termination fine-tuning repairs cross-stage causal effects”.
>   - ReinforceGen as a standalone high-performing visuomotor policy for multi-stage robot manipulation tasks is also a valuable contribution. Multiple works [4, 5] have demonstrated the potential of directly deploying a vision-based RL policy in the real world. Additionally, designing low-dimensional privileged information is not always trivial, such as when working with deformable objects.
>
> *“... apply MimicGen, BC, and residual RL for each individual subtask in simulation, then alternately execute motion planning and these policies for distillation.”*
> - Training each subtask individually has the following issues: 1) The initiation distribution of the next stage is dictated by the execution of the previous one. This includes joint configurations and grasping positions, which are not trivial to generate in individual training. 2) As mentioned in Section 4.3 “Termination fine-tuning” and Figure 3C, even human-engineered termination signals can be imperfect and lead to reward hacking. ReinforceGen alleviates this issue by purging predicted terminations with low final task success rates. This is dependent on future stage policies - not accessible in individual training.
>
> *“... comparison of previous works solving the similar problem”*
> - We appreciate the reviewer for pointing out the relevance of these works. However, even though they solve a similar problem to ours, all three works make stronger assumptions than our setting, which makes the comparison unfair. Specifically, [1] used engineered dense rewards for RL training, contrary to our 0-1 reward assumption, as well as using privileged object state information in observations. [2] also used object state information for skill training and planning, and stated in Appendix A-Q3 that their method can face challenges when generalizing to high-dimensional input such as images. [3] used engineered dense rewards for RL training and object state information in observations.
>
> *“... sim-to-real performance and real-world applicability”*
> - We acknowledge the lack of real-world experiments. In light of this, we propose a sim-to-real transfer procedure for ReinforceGen and argue that simulation experiments are also valuable. We kindly refer the reviewer to our common response for a detailed discussion.
>
> *“... synthesizing more data (e.g. 10000 demos)”*
> - Referencing Appendix E in [6], scaling the generated demos from 1000 to 5000 does increase the performance of HSP policies in some of the tasks. Therefore, the improvement of ReinforceGen can decrease due to the base policy being better. However, we would like to point out: 1) The performance increase from increasing generated demos is not universal and can be limited depending on the task; 2) Significantly increasing the generated demos incurs extra cost in generation and imitation learning.
>
> *“Why is DrQ-v2 used for residual RL?”*
> - We follow previous work [7] for the choice of the RL algorithm. Other RL algorithms, such as PPO and SAC, may also reach similar performances, but we consider it orthogonal to our work. The design of ReinforceGen enables the user to select different off-the-shelf RL algorithms that are best suited for the task.

---

> ### Author Response · Authors · 2025-11-26
>
> ## References
> [1] Chen et al., Sequential Dexterity: Chaining Dexterous Policies for Long-Horizon Manipulation, 2023
>
> [2] Agia et al., STAP: Sequencing Task-Agnostic Policies, 2023
>
> [3] Lee et al., Adversarial Skill Chaining for Long-Horizon Robot Manipulation via Terminal State Regularization, 2021
>
> [4] Ankile et al., From Imitation to Refinement -- Residual RL for Precise Assembly, 2024
>
> [5] Yuan et al., Learning to Manipulate Anywhere: A Visual Generalizable Framework For Reinforcement Learning, 2024
>
> [6] Garrett et al., Skillmimicgen: Automated demonstration generation for efficient skill learning and deployment, 2024
>
> [7] Zhou et al., SPIRE: Synergistic Planning, Imitation, and Reinforcement Learning for Long-Horizon Manipulation, 2024

---

### Author Response · Authors · 2025-11-26
**General Response to Reviewers**

## Real World Evaluations
We acknowledge that multiple reviewers have raised concerns about the lack of real-world evaluations. We want to address this issue from the following aspects.
- **ReinforceGen can be used in conjunction with existing sim-to-real strategies.** We propose the following sim-to-real deployment procedure and argue that it can work based on evidence from prior works:
  - We start from a simple zero-shot deployment following [1]: At the beginning of each stage, we use FoundationPose [2] to estimate the object poses and map the result to a simulator. We then execute ReinforceGen in simulation and replay the action sequence in the real world.
  - We can then add the samples collected from the previous step to the training set of the initiation pose predictor (Section 4.1) and the replay buffer in RL finetuning (Section 4.2) to establish a procedure similar to co-training[3, 4]. This step enables us to adapt our visuomotor policy to real-world visual and dynamics. We can then execute the adapted policy on a real robot. If the result is still not satisfactory, we then repeat this step with the newly collected data.
- **Simulation experiments provide insight into what will happen in the real world but with larger number of trials.**
  - Multiple techniques used in ReinforceGen, including target pose replanning (Section 4.1) and termination fine-tuning (Section 4.3), are all applicable in the real world.
  - Several prior works have also conducted investigations in simulation and shown that the results transfer to real-world settings. For example, [5, 6] investigated the design choices in learning from offline demonstrations in sim and found their conclusions applicable to real-world manipulation tasks.

## Paper Revisions
We have made several changes to our paper to reflect the comment from the reviewers. The updated parts are marked blue in our revision. The detailed update list is as follows:
- L252 and Figure 3: Removed the inaccurate usage of “distillation”.
- Figure 4 and Appendix E.1: Added missing details for this experiment.
- Section 4.4: Updated the wording of this section to make it self-contained.
- Appendix B.1: Added missing details for observations.
- Appendix C.2 and E.2: Added missing implementation details.

## References
[1] Garrett et al., Skillmimicgen: Automated demonstration generation for efficient skill learning and deployment, 2024

[2] Wen et al., FoundationPose: Unified 6D Pose Estimation and Tracking of Novel Objects, 2023

[3] Ankile et al., From Imitation to Refinement -- Residual RL for Precise Assembly, 2024

[4] Maddukuri et al., Sim-and-Real Co-Training: A Simple Recipe for Vision-Based Robotic Manipulation, 2025

[5] Mandlekar et al., What Matters in Learning from Offline Human Demonstrations for Robot Manipulation, 2021

[6] Saxena et al., What Matters in Learning from Large-Scale Datasets for Robot Manipulation, 2025

---

### Author Response · Authors · 2025-12-04

We thank the reviewers for their insightful evaluations and feedback. We have provided responses to all reviewer comments and believe these sufficiently address their concerns. We are encouraged by the reviewers’ general consensus on the competitive performance and demo-efficiency of ReinforceGen in simulation. Regarding the absence of real-world experiments, we would like to reiterate the following points from our previous general response:
- We have proposed a real-world deployment pipeline that is expected to work, supported by evidence from prior literature;
- Simulation experiments offer important insights into real-world performance and therefore hold scientific value.

---

### Meta-Review · Area_Chair_27E4 · 2026-01-11

**Summary:**

The four reviewers collectively raised several significant concerns about this submission. The most prominent and universally cited weakness is the absence of real-world experiments, which all reviewers identified as a critical gap for a robotics paper. Beyond this central issue, reviewers questioned the scientific novelty of the work, with Reviewer GL2P characterizing it as "engineering work" that combines existing components without clear scientific contributions. Additional concerns included the narrow evaluation scope (limited to five Robosuite tasks), missing comparisons to relevant prior work on skill chaining, incomplete implementation details, and questions about design choices such as the use of imitation learning versus pure RL and the selection of DrQ-v2 as the RL algorithm.

**Reviewer Concerns:**

Concerns adequately addressed:

The authors successfully addressed several technical and presentation issues. They clarified the design rationale for using imitation learning over pure RL by referencing SPIRE's findings that pure RL struggles with sparse rewards in this domain. They explained the choice of DrQ-v2 as following prior work conventions. The manuscript revisions corrected the inappropriate use of "distillation," added missing implementation details (thresholds, reset distributions, observation specifications), and improved Section 4.4 to be self-contained. The authors also provided reasonable responses regarding comparisons to related work, noting that the suggested baselines (Sequential Dexterity, STAP, Adversarial Skill Chaining) make stronger assumptions that would render comparisons unfair.

Concerns not adequately addressed:

The fundamental concern regarding real-world validation remains outstanding. The authors proposed a theoretically plausible sim-to-real pipeline combining FoundationPose for pose estimation with co-training procedures, but this pipeline was not implemented or validated. As Reviewer 1Aqh explicitly noted, comparable works such as MimicGen and DexMimicGen include real-robot experiments, making this omission particularly conspicuous. The argument that "simulation experiments offer important insights into real-world performance" does not adequately address concerns about simulation-specific artifacts that may not transfer to hardware. The limited benchmark diversity (single simulator, five tasks) also remains unaddressed beyond assertions that these tasks are representative.

**Reviewer Scores:**

Reviewer NVAj (Initial: 4): This reviewer did not respond to the rebuttal. Based on the nature of their concerns - particularly the suggestion that a simpler privileged baseline might work equally well and the emphasis on sim-to-real evaluation - I project they would maintain their score of 4. The authors provided reasonable counterarguments but did not present compelling new evidence.

Reviewer 7A7r (Initial: 4): This reviewer explicitly stated that the authors "did not address the main issues" and maintained their score. The projected final score remains 4.

Reviewer GL2P (Initial: 2): This reviewer's concerns centered on lack of scientific contribution and missing real-world experiments. The authors' response defending the engineering integration as scientific contribution is unlikely to change this reviewer's fundamental assessment. Neither concern was addressed with new experiments or analysis. Projected final score is 2.

Reviewer 1Aqh (Initial: 4): This reviewer explicitly acknowledged the rebuttal but stated that the main concern regarding real-world evaluation remains, noting that "demonstrations on real hardware are extremely important" and "there are too many tricks in simulators." Maintained the score of 4.

Based on the reviews, rebuttal, and discussion, I recommend rejection of this paper. The core contribution is a competent integration of existing techniques (MimicGen-style data generation, residual RL, termination gating) that achieves strong demonstration efficiency. However, the unanimous concern about missing real-world validation is substantial for a robotics venue, particularly when closely related works have established this as a standard expectation. The proposed but unimplemented sim-to-real pipeline does not mitigate this concern. The final reviewer consensus (three scores of 4, one score of 2) reflects a paper that falls below the acceptance threshold, and the discussion did not produce evidence that would shift this assessment. The paper would benefit from either real-world experiments or significantly expanded simulation studies demonstrating robustness to sim-to-real gaps before resubmission.

---

### Decision · Program_Chairs · 2026-01-26

Reject